# Significant Effects of Long-Term Application of Straw and Manure Combined with NPK Fertilizers on Olsen P and PAC in Red Soil

**Fengxia Sun [1,2,3], Nan Sun [3,*], Boren Wang [3], Zejiang Cai [3] and Minggang Xu [2,3,*]**

1    College of Tropical Crops, Hainan University, Haikou 570228, China; sfx28@foxmail.com
2    Shanxi Province Key Laboratory of Soil Environment and Nutrient Resources, Institute of Eco-Environment and Industrial Technology, Shanxi Agricultural University, Taiyuan 030031, China
3    State Key Laboratory of Efficient Utilization of Arid and Semiarid Arable Land in Northern China, Key Laboratory of Arable Land Quality Monitoring and Evaluation, Ministry of Agriculture and Rural Affairs/Institute of Agricultural Resources and Regional Planning, Chinese Academy of Agricultural Sciences, Beijing 100081, China; wangboren@caas.cn (B.W.); caizejiang@caas.cn (Z.C.)
*    Correspondence: sunnan@caas.cn (N.S.); xuminggang@caas.cn (M.X.)

**Abstract:** The application of manure (M) and straw (S) will increase the Olsen P and phosphorus activation coefficient (PAC) in soil. Clarifying the increasing trend of Olsen P and PAC is crucial for rational fertilization. This study fitted the equation between the accumulated P surplus, Olsen P, and PAC in four treatments for 28 years and analyzed the changes and rates of P fractions. The results showed Olsen P and PAC increase linearly with NPK and NPKS treatments; for every 100 kg ha$^{-1}$ of P surplus, Olsen P increased by 5.9 and 6.7 mg kg$^{-1}$, and PAC increased by 0.52% and 0.50%. With M and MNPK treatments, the sigmoid curve equation was the best fitting method. The equilibrium values were 167 and 164 mg kg$^{-1}$ for Olsen P, and 10.4 and 10.2 mg kg$^{-1}$ for PAC. There was a correlation between Al-P, Ca$_2$-P, Resin-P, NaOH-P$_i$, C/N, SOC, and pH, which had the highest interpretation rates for Olsen P and PAC. Manure is significantly better than straw in improving Olsen P in red soil. It is recommended to reduce the amount of manure applied for a long time to avoid a zero increase in Olsen P.

**Keywords:** red soil; manure; straw; Olsen P; phosphorus activation coefficient

## 1. Introduction

Long term application of P fertilizers combined with manure or straw is a commonly used fertilization method in soil, which could continuously increase the Olsen P and PAC [1–3]. Olsen P is the most effective for crops, but it is prone to loss, especially in the red soil areas with high rainfall in Southern China [3,4]. Therefore, elucidating the differences in Olsen P and PAC under P surplus after applying manure and straw is crucial for selecting a reasonable fertilization method. The input fertilizers interact with P fractions and soil properties; therefore, the P fractions and the effects of soil properties also need further clarification. However, after 28 years of long-term fertilization, the changes and influencing factors of Olsen P, PAC, and P fractions in red soil are still unclear. Based on its importance in guiding red soil fertilization, further analysis and research are needed. In order to elucidate the dynamic changes of Olsen P and PAC under the addition of manure and straw in red soil, researchers fitted the relationship between P budget and the increase of Olsen P (ΔOlsen P) and PAC (ΔPAC), and long-term experiments provided a good research platform [5–7]. In previous studies, with the surplus of soil P as the abscissa and ΔOlsen P as the ordinate, the ΔOlsen P per 100 kg ha$^{-1}$ of surplus was calculated by using the linear equation fitting so as to predict and evaluate the input of P fertilizer [8]. Ahmed and Suriyagoda's research showed that NPKM treatment increased TP and Olsen P, and returning straw to the field improved crop root growth and significantly increased

grain yield, which is a better fertilization method [9,10]. Amanullah and Zhuang's research also showed that the application of manure and straw increases crop productivity, and partially replacing fertilizers with straw was an effective measure to achieve crop waste utilization and reduce greenhouse gas emissions [11,12]. However, Whalen's research showed that long-term application of manure could increase P loss in irrigation areas and increased the risk of total P pollution in groundwater [13]. Hua's research showed that long-term, continuous, and excessive use of farmyard manure to increase crop yields was not sustainable and had a high risk of P environmental pollution. The MNPK treatment significantly increased soil Olsen P; however, the NPKS treatment had no significant effect on Olsen P. The average increase in PAC and Olsen P in manure was significantly higher than straw [14]. However, for red soil, the impact of long-term application of straw and manure on Olsen P and PAC is still unclear; we suspect that its increase is phased. Therefore, on the basis of previous studies, in order to clarify the effects of different carbon source additives (manure and straw) and fertilizers on P, we reassessed ΔOlsen P and ΔPAC in red soil under long-term P input.

Important factors for the low utilization rate of P fertilizer in red soil is the fixation of P by iron and aluminum oxides. Fei, Koch, and Stutter's research showed that the migration of P was controlled by its fractions; the absorption of P by crops was affected by the fixation of P by iron oxide and aluminum oxide, and improving agricultural efficiency requires clarifying iron and aluminum oxides in the soil [15–17]. Thus, it is important to clarify the dynamic changes of iron and aluminum oxides under different fertilization methods in red soil. Jiang-Gu and Hedley's methods were widely used to evaluate the P fractions, which could clarify the changes of the P pool [18–20]. Analyzing the relationship between ΔOlsen P, ΔPAC. and P fractions can clarify the impact of the P pool, and provide data support for adjusting fertilization. The main factors affecting Olsen P were soil physical and chemical properties, and the fertilizer; Olsen P is influenced by the type and amount of fertilizer input, soil organic matter, and pH [21–23]. Long-term application of nitrogen fertilizer can significantly reduce the pH of red soil. Ahmed and Abdala showed that the application of chicken manure changed the pH and reduced the concentration of exchangeable aluminum; thus, the soil pH affected the unstable P pool [4,9]. Debicka and Yang showed that SOC played a major role in the process of P adsorption; it increased the maximum adsorption capacity of P, which improved the effectiveness of P by reducing the adsorption strength of P and the maximum phosphate release capacity [24,25]. However, interaction between soil physical properties, chemical properties, Olsen P, and PAC with different fertilizers is unclear in red soil. Therefore, the change trends and influence factors of Olsen P and PAC on accumulated P need to be further evaluated in red soil. The purpose of this paper was to systematically and comprehensively evaluate the change in trends and influencing factors of Olsen P and PAC in soil accumulated phosphorus (P) under different fertilization methods.

To date, regarding the long-term application of manure and straw in red soil, research on Olsen P and PAC change based on fertilization is still weak. Therefore, clarifying the Olsen P, PAC, and identifying the influencing factors will provide data support for optimizing the application of phosphate fertilizer, and provide reference for formulating reasonable laws and regulations on soil P application. The objectives of this study were as follows: first, to reevaluate the growth trend of Olsen P and PAC in accumulated P in red soil after 28 years of application of manure and straw; second, to study on the interaction of P fractions (Jiang-Gu and Hedley methods) and soil chemical properties with PAC and Olsen P; and third, to provide a reference for adjusting the amount of phosphate fertilizer in red soil areas.

## 2. Materials and Methods

### 2.1. Site Description and Used Material

The fertilization experiment was carried out in the Qiyang (QY) Red Soil Experimental Station, located in Hunan Province in China. The experimental field site was established

in 1990 under a wheat maize rotation system. Detailed information on geographical location, climate, and the initial physical and chemical properties is given in Table 1. We selected four treatments, including nitrogen, phosphorus, and potassium (NPK); nitrogen, phosphorus, potassium, and straw (NPKS); manure (M); and manure, nitrogen, phosphorus, and potassium (MNPK). Fertilizer inputs for each treatment are shown in Table 2. An analysis was conducted on the composition of manure and straw from three fertilization treatments (NPKS, M, and MNPK) in 1991, 2004, 2008, and 2012, including the annual input, water content, and nutrient content of manure and straw. The carbon, nitrogen, phosphorus, and potassium contents of manure and straw were shown in Supplementary Table S1.

After 28 years of long-term fertilization, there were significant changes in soil chemical properties compared to the initial soil. For NPKS treatment, TP increased from 0.45 g kg$^{-1}$ to 1.13 g kg$^{-1}$, and SOC increased from 7.9 g kg$^{-1}$ to 10.4 g kg$^{-1}$. The increase in TP and SOC in M and MNPK treatments was significantly higher than that in NPKS treatment. The NPKS treatment significantly reduced the soil pH from 5.7 to 4.3, while the MNPK treatment increased it to 5.91 and the M treatment was 6.73. The soil physical and chemical properties of the four treatments after 28 years of different fertilization were shown in Supplementary Table S4.

**Table 1.** Geographical location, climate, physical properties, and chemical properties (1990) of the QY long-term fertilizer experiment.

| Latitude | Longitude | Altitude (m) | MAT (°C) | MAP (mm) | MAE (mm) |
|---|---|---|---|---|---|
| 26°45′ | 111°52′ | 120 | 18.0 | 1426 | 1435 |
| SOC (g kg$^{-1}$) | TN (g kg$^{-1}$) | AN (mg kg$^{-1}$) | TP (g kg$^{-1}$) | Olsen-P$_0$ (mg kg$^{-1}$) | TK (g kg$^{-1}$) |
| 7.9 | 1.07 | 79 | 0.45 | 13.9 | 13.7 |
| AK (mg kg$^{-1}$) | pH | Sand (%) | Silt (%) | Clay (%) | BD (g cm$^{-3}$) |
| 104 | 5.7 | 24.3 | 31.8 | 43.9 | 1.2 |

MAT is the mean annual temperature, MAP is the mean annual precipitation, MAE is the mean annual evaporation, SOC is soil organic carbon, TN is total nitrogen, AN is available nitrogen, TP is total phosphorus, Olsen P$_0$ is available P, TK is total potassium, AK is available potassium, BD is bulk density.

**Table 2.** Information on the long-term fertilizer treatments.

| Treatment | Inorganic Fertilizer (N-P-K kg ha$^{-1}$ yr$^{-1}$) | | Manure (t ha$^{-1}$ yr$^{-1}$) | Types of Manure | Straw | Crop Rotation |
|---|---|---|---|---|---|---|
| NPK [1] | C [2]-210-36.7-69.7 | W-90-15.7-30 | 0 | - | 0 | |
| NPKS | C-210-36.7-69.7 | W-90-15.7-30 | 0 | - | 1/2 straw returning to field | Wheat-Corn |
| M | C-0-0-0 | W-0-0-0 | C-42/W-18 | Pig manure | 0 | |
| MNPK | C-63-36.7-69.7 | W-27-15.7-30 | C-29.4/W-12.6 | Pig manure | 0 | |

[1] NPK, nitrogen, phosphorus, potassium; NPKS, nitrogen, phosphorus, potassium, straw; M, manure; MNPK, manure, nitrogen, phosphorus, potassium. [2] Crops were corn (C) and wheat (W).

## 2.2. Experimental Design and Field Management

The experiment was randomized, and each treatment was repeated twice (plot size 196 m$^2$). Calcium superphosphate was used as a source of chemical P fertilizer, and urea and potassium chloride were nitrogen and potassium fertilizers. All fertilizers were applied to the soil together. Pig manure was obtained locally every year with an average dry matter content of 30%. The average content of P in dried manure was 1.0%, carbon was 40%, nitrogen was 1.6%, and potassium was 1.5%. With NPK and NKPS treatments, a total

of 52.4 kg ha$^{-1}$ of fertilizer P was applied to wheat and maize every year. With MNPK treatment, 42 t ha$^{-1}$ yr$^{-1}$ was applied to fresh pig manure. With M treatment, application rate of fresh pig manure was 60 t ha$^{-1}$ each year, with a P input of 180 kg ha$^{-1}$. Adopting a double cropping rotation system of wheat and corn, the fertilizer application in the corn season accounts for 70% of the manure application, while wheat accounts for 30%. The fertilizer is applied as a base fertilizer before sowing wheat and corn. Except for NPKS treatment, where half of the crop straw is returned to the field, all the aboveground parts of the other treated crops are taken away. The yield of wheat and corn from the initial period of 1990 to 2018 is shown in Supplementary Table S3. Before sowing, chemical fertilizer was applied to the soil surface in the form of turning over and pressing, and the tillage depth was about 20–25 cm. Half of the stalks of NPKS-treated crops were returned to the field, and the rest of the aboveground crops were taken away.

Crops were not irrigated in all treatments; 4.5 kg ha$^{-1}$ of carbofuran was sprayed when the corn borer was controlled at the filling/bell stage of corn, 3.0 kg ha$^{-1}$ of carbendazim insecticide was sprayed at the jointing stage of wheat, and 3.75 kg ha$^{-1}$ of omethoate insecticide was sprayed at the filling stage. All treated weeds at the test site were removed by hand, and all aboveground biomass was harvested manually, and then separated into grain and straw. The test samples were dried to constant weight through the oven.

### 2.3. Soil Sampling and Analysis

Soil samples (0–20 cm) were collected after harvesting annually at the Qiyang experiment site, and there were five cores for the treatments. All tested drugs are purchased from China National Pharmaceutical Chemical Reagent Co., Ltd. (Shanghai). The soil Olsen P was determined by the Olsen method (1954), which was extracted using $NaHCO_3$ (0.5 mol L$^{-1}$,). Refer to Lu (2000) and Page (1982) for other soil physical and chemical properties [26,27]. Total phosphorus (TP) was digested with $H_2SO_4$-$HClO_4$ and measured using the molybdenum-blue colorimetric method. One of the fractions of P was determined by the Jiang-Gu method [19]. This method uses mixed extractant to extract ferric phosphate. Specifically, 0.25 mol L$^{-1}$ $NaH_2CO_3$ is used to extract dicalcium phosphate (Ca$_2$-P); 1 mol L$^{-1}$ $NH_4Ac$ leaching octacalcium phosphate (Ca$_8$-P); 0.3 mol L$^{-1}$ sodium citrate $Na_2S_2O_4$-NaOH leaching closed storage P (O-P); 0.5 mol L$^{-1}$ $H_2SO_4$ leaching decacalcium phosphate (Ca$_{10}$-P); 0.l mol L$^{-1}$ NaOH-$Na_2CO_3$ leaching iron phosphate (Fe-P); and 0.5 mol L$^{-1}$ $NH_4F$ leaching aluminum phosphate (Al-P). We corrected the P fractions with the TP to avoid the systematic errors. Another method was the Tiessen modified Hedley method [19,20]. P was initially extracted with an anionic-exchange resin (resin-P) and then with $NaHCO_3$ ($NaHCO_3$-P). Resin-P and $NaHCO_3$-P are assumed to be labile P fractions. Moderately labile P sorbed on amorphorus Fe and Al, and minerals were subsequently extracted with NaOH (NaOH-P), followed by ultrasonification in NaOH to obtain "protected P" occluded or contained within aggregates. Primary mineral P was extracted with HCl (HCl-P), and the remaining P was removed by an $H_2O_2$-$H_2SO_4$ digestion (Residual-P).

### 2.4. Data Analysis

The calculation formula for P surplus and fitting equation is as follows:

$$\Delta \text{Olsen P} = \text{Olsen P}_t - \text{Olsen P}_{1990} \tag{1}$$

where Olsen P$_t$ and Olsen P$_{1990}$ are Olsen P (mg kg$^{-1}$) at the t$^{th}$ and initial year.

$$\text{PAC}_t\% = \text{Olsen P}_t \times 100/\text{TP}_t \tag{2}$$

where TP$_t$ (mg kg$^{-1}$) is total phosphorus at the t$^{th}$ year.

$$\Delta \text{PAC}\% = \text{PAC}_t\% - \text{PAC}_{1990}\% \tag{3}$$

where $PAC_t$ is PAC (%) at the $t^{th}$ year and $PAC_{1990}$ is PAC (%) at initial year.

$$P_{input} = P_{CP} + P_{MP} \tag{4}$$

where $P_{input}$ (kg ha$^{-1}$) is phosphorus applied (manure and fertilizer phosphate) every year, $P_{CP}$ and $P_{MP}$ (kg ha$^{-1}$) are phosphorus applied in fertilizer and applied in manure annually.

$$P_{uptake} = P_G \times Y_G + P_S \times Y_S \tag{5}$$

where $P_{uptake}$ (kg ha$^{-1}$) is the TP absorbed by the crop after harvest (grain and straw), $P_G$ and $P_S$ are the phosphorus content in grain and straw, $Y_G$ and $Y_S$ are grain and straw yield.

$$P_{surplus} = \sum_{1990}^{yr}(P_{input} - P_{uptake}) \tag{6}$$

where $P_{surplus}$ (kg ha$^{-1}$) is the accumulative P surplus.

### 2.5. Statistical Analyses

The data of red soil for 28 years were selected in the long-term experiment. The abscissa represents the annual accumulated P surplus, and the ordinate represents the annual increase of Olsen P. The relationship between ΔOlsen P and P surplus was examined using the fitting relationship linear equation for NPK and NPKS treatments, but S-shaped curves for M and MNPK treatments. The data of each measurement variable were analyzed by one-way ANOVA, and the least significant difference (LSD) at $p = 0.05$ was used to compare the significant differences between the treatments. Redundancy analysis (RDA) was used to analyze the influencing factors of Olsen P and PAC, which is a widely used data analysis tool and can determine the correlation for Olsen P and PAC and other parameters. The RDA analysis adopted Canoco—version 5 software. All the diagrams were drawn using sigmaplot—Version 12.5 and Excel—2013.

## 3. Results

### 3.1. Increase of Olsen P under P Surplus

The accumulated P surplus showed an increasing trend in red soil for four treatments (Figure 1). The increase range of accumulated P among the four treatments ranked as M > MNPK > NPS ≈ NPK. P application increased accumulated P, and after 28-year of long-term fertilization, the accumulated P surplus was 4203 and 3901 kg ha$^{-1}$ under M and MNPK treatments, and 970 and 926 kg ha$^{-2}$ under NPK and NPKS treatments. In conclusion, the accumulated P of NPK and NPKS treatments was significantly lower than that of M and MNPK treatments. Compared to NPKS treatment, M and MNPK treatment significantly increased the accumulation of P in red soil ($p < 0.05$).

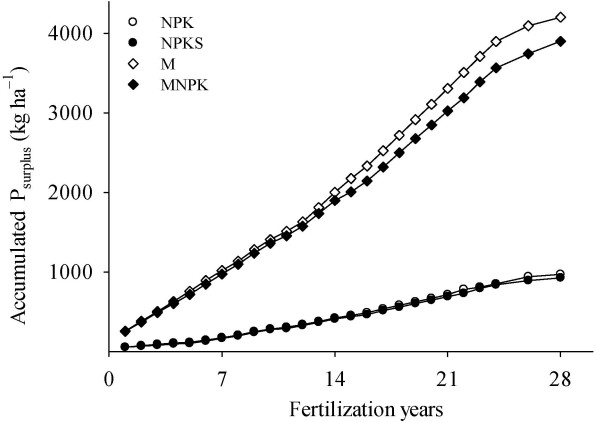

**Figure 1.** Accumulation of P surplus for 28 years. NPK, nitrogen, phosphorus, potassium; NPKS, nitrogen, phosphorus, potassium, straw; M, manure; MNPK, manure, nitrogen, phosphorus, potassium.

In the case of surplus soil P, ΔOlsen P in four fertilizer treatments was analyzed, and two fitting equations were used. Olsen P showed a linear increase trend under NPK and NPKS treatments (Figure 2a). A comparison of the slopes $S_{NPK}$ and $S_{NPKS}$ of the linear equation showed an order of $S_{NPKS}$ (0.067) > $S_{NPK}$ (0.059), and for every 100 kg ha$^{-1}$ of P surplus, Olsen P increased 6.3 mg kg$^{-1}$. Under the treatments of M and MNPK, the fitting equation of Olsen P was more in line with sigmoid equation (Figure 2b). The equilibrium value ($E_M$, $E_{MNPK}$) of ΔOlsen P was 167 and 164 mg kg$^{-1}$ for M and MNPK treatments. Application of manure and P fertilizer led to increase of Olsen P. In conclusion, compared to NPKS treatment, M and MNPK treatment significantly increased Olsen P in red soil ($p < 0.05$).

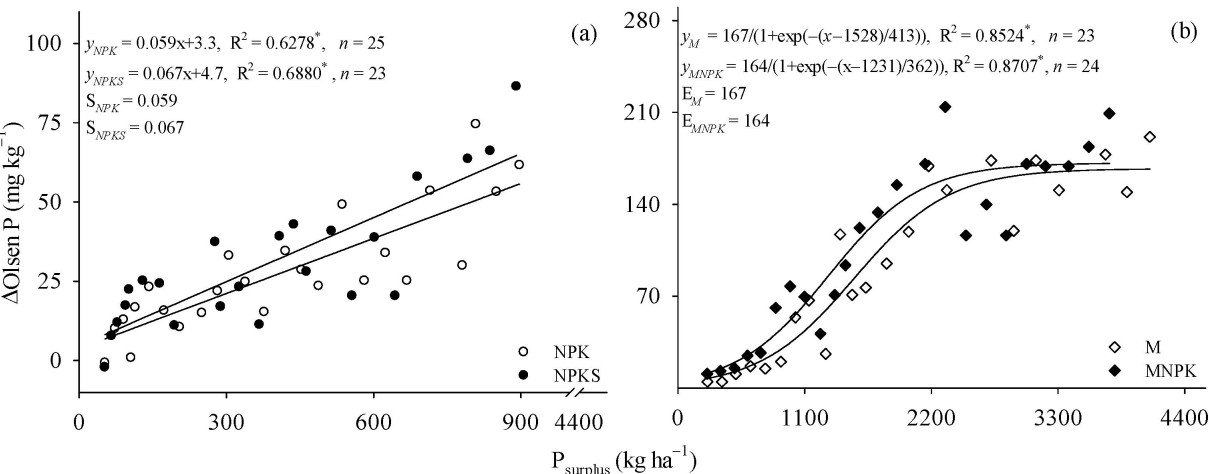

**Figure 2.** Relationship between ΔOlsen P and different amounts of P surplus. NPK, nitrogen, phosphorus, potassium; NPKS, nitrogen, phosphorus potassium, straw; M, manure; MNPK, manure, nitrogen, phosphorus, potassium. $S_{NPK}$, the slope of the fitting equation by NPK treatment, $S_{NPKS}$, the slope of the fitting equation by NPKS treatment (**a**). Fitted equation was a sigmoidal equation of $y_{M/MNPK} = a/(1 + \exp(-(x - x_0)/b))$, where "a" represents the equilibrium value, *, $p < 0.05$. $E_M$ is the equilibrium value by M treatment. $E_{MNPK}$ is the equilibrium value by MNPK treatment (**b**).

### 3.2. Increase of PAC under P Surplus

In the case of surplus soil P, the increasing trend of PAC in four fertilizer treatments was analyzed, and two fitting equations were used to fit it. ΔPAC showed a linear increasing trend under NPK and NPKS treatments (Figure 3). The slopes $S_{NPK}$ and $S_{NPKS}$ of the linear equation were 0.0052 and 0.0050, respectively. PAC increased by 0.51% on average for 100 kg ha$^{-1}$ of P surplus. Under M and MNPK treatments, the sigmoid curve equation was the best fitting method, and when the soil accumulated P reached an average of 1660 kg ha$^{-1}$, the increase of PAC remained unchanged. The equilibrium values of $E_M$ and $E_{MNPK}$ of the M and MNPK treatments were similar too, and the equilibrium value of ΔPAC was 10.4% and 10.2% for M and MNPK treatments. In conclusion, there was no significant difference in the increase of PAC between NPK and NPKS treatments. Compared with NPK treatment, the application of straw did not significantly increase PAC. Compared to NPKS treatment, M and MNPK treatment significantly increased the PAC of red soil ($p < 0.05$).

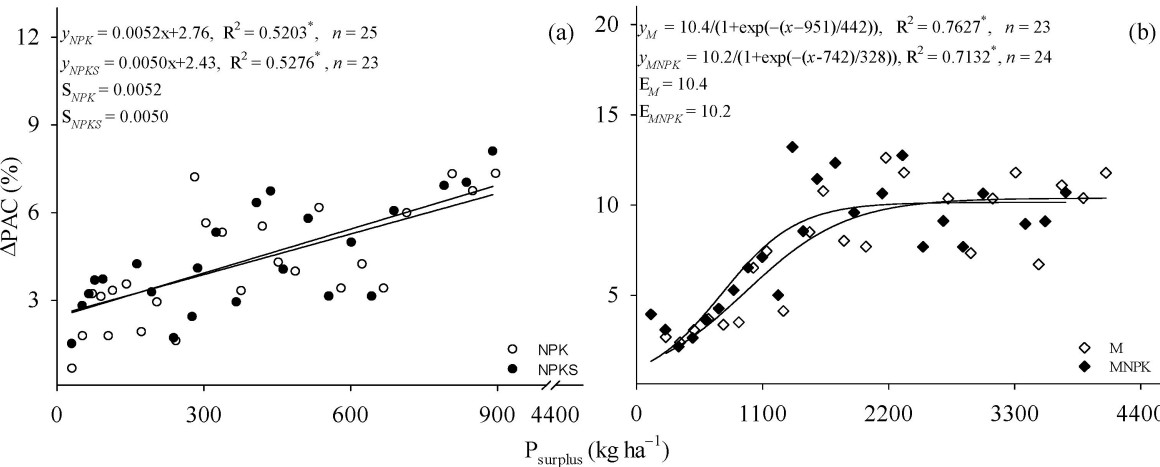

**Figure 3.** Relationship between PAC and different amounts of P surplus. PAC, phosphorus activation coefficient. NPK, nitrogen, phosphorus, potassium; NPKS, nitrogen, phosphorus potassium, straw; M, manure; MNPK, manure, nitrogen, phosphorus, potassium. *, $p < 0.05$. $S_{NPK}$, the slope of the fitting equation by NPK treatment, $S_{NPKS}$, the slope of the fitting equation by NPKS treatment (**a**), $E_M$ is the equilibrium value by M treatment. $E_{MNPK}$ is the equilibrium value by MNPK treatment (**b**).

### 3.3. Comparison of P Fraction Changes for Jiang-Gu Method

The increase of P fractions in 2000 and 2013 was analyzed (Table 4). In order to compare the increase of various P fractions for the Jiang-Gu method, the P fractions in 1990 were subtracted from that in two years (2000 and 2013), respectively. Compared with all the P fractions at the beginning of the experiment in 1990, Al-P of all treatments between 2000 and 2013 increased continuously, while Fe-P increased rapidly at first and then slowly.

After 10 years of fertilization (2000), the total Ca-P ($Ca_2$-P + $Ca_8$-P + $Ca_{10}$-P) increased by 13.1 mg $kg^{-1}$ for NPKS treatment, and the MNPK treatment increased it by 154.7 mg $kg^{-1}$, which was 11.8 times higher than the NPKS treatment. NPKS treatment reduced the Ca-P ratio by 6.9%, while MNPK treatment increased it by 1.3%. Similarly, the increase in aluminum P ($\Delta$Al-P) and iron P ($\Delta$Fe-P) with MNPK treatment was 2.3 and 2.0 times higher than that with NPKS treatment. After 23 years of fertilization (2013 year), the increase in Ca-P was significantly higher than in 2000, the total Ca-P increased by 213.6 mg $kg^{-1}$ for NPKS treatment, and the MNPK treatment increased it by 532.6 mg $kg^{-1}$, which was 2.5 times higher than the NPKS treatment. In summary, after 23 years of continuous fertilization, the increase in calcium P and aluminum P was significantly higher than that of 10 years of fertilization ($p < 0.05$). The increase in P fractions in the treatment of manure and manure combination NPK application was higher than that in the treatment of straw combination NPK fertilization.

### 3.4. Comparison of P Fraction Changes for Hedley Method

The increase of P fractions for the Hedley method in 2000 and 2013 was analyzed (Table 3); the P fractions in 1990 were subtracted from that in two years (2000 and 2013), respectively. After 23 years of continuous fertilization, the increase in inorganic P was significantly higher than that of 10 years of fertilization ($p < 0.05$). Similar to the Jiang-Gu method, the increase in P fractions in the treatment of manure and manure combination NPK application was higher than straw combination NPK fertilization treatment.

**Table 3.** Increase in the amount and percentage of P fractions for the Hedley Method.

| Treatment | ΔResin-P [1] (mg kg$^{-1}$) | ΔNaHCO$_3$-P$_i$ (mg kg$^{-1}$) | ΔNaHCO$_3$-P$_o$ (mg kg$^{-1}$) | ΔNaOH-P$_i$ (mg kg$^{-1}$) | ΔNaOH-P$_o$ (mg kg$^{-1}$) | ΔDHCl-P$_i$ (mg kg$^{-1}$) | ΔCHCl-P$_i$ (mg kg$^{-1}$) | ΔCHCl-P$_o$ (mg kg$^{-1}$) | Δresidual-P (mg kg$^{-1}$) | Δresin-P (mg kg$^{-1}$) | ΔNaHCO$_3$-P$_i$ (mg kg$^{-1}$) | ΔNaHCO$_3$-P$_o$ (mg kg$^{-1}$) | ΔNaOH-P$_i$ (mg kg$^{-1}$) | ΔNaOH-P$_o$ (mg kg$^{-1}$) | ΔDHCl-P$_i$ (mg kg$^{-1}$) | ΔCHCl-P$_i$ (mg kg$^{-1}$) | ΔCHCl-P$_o$ (mg kg$^{-1}$) | Δresidual-P (mg kg$^{-1}$) |
|---|---|---|---|---|---|---|---|---|---|---|---|---|---|---|---|---|---|---|
| | | | | | 2000 (year) | | | | | | | | | 2013 (year) | | | | |
| NPK[2] | 3.6 ± 1.0 bB | 54.3 ± 6.6 cB | 4.5 ± 1.6 cB | 81.2 ± 10.9 cB | 1.4 ± 0.6 cB | 4.3 ± 0.7 dB | 31.6 ± 4.4 bB | 9.1 ± 1.8 aB | 3.6 ± 0.5 cB | 99.5 ± 12.8 aA | 116.8 ± 21.5 cA | 8.7 ± 2.8 bA | 133.9 ± 21.8 cA | 12.8 ± 3.3 bA | 57.0 ± 6.5 cA | 103.1 ± 8.9 bA | 17.5 ± 3.6 bA | 11.6 ± 2.8 cA |
| NPKS | 20.8 ± 2.9 aB | 44.3 ± 6.5 cB | 11.8 ± 2.1 bA | 112.6 ± 9.8 bB | 1.4 ± 0.4 cB | 10.0 ± 1.2 cB | 63.2 ± 10.0 aB | 10.0 ± 2.1 aB | 20.0 ± 5.3 bA | 107.7 ± 10.8 aA | 143.7 ± 9.9 cA | 9.0 ± 2.2 bA | 176.7 ± 20.6 bA | 2.6 ± 0.6 cA | 55.6 ± 8.9 cA | 121.4 ± 11.5 abA | 21.6 ± 5.5 abA | 15.0 ± 3.5 cA |
| M | 31.6 ± 12.1 aB | 78.6 ± 9.0 bB | 22.9 ± 4.2 aA | 182.1 ± 31.7 aB | 3.8 ± 1.0 bB | 48.3 ± 7.2 bB | 68.1 ± 5.9 aB | 11.1 ± 2.9 aB | 31.6 ± 6.8 aB | 98.5 ± 7.5 aA | 199.6 ± 26.6 bA | 15.8 ± 2.5 aA | 366.0 ± 44.5 aA | 21.4 ± 6.7 abA | 98.2 ± 9.0 bA | 119.6 ± 20.0 abA | 23.4 ± 3.8 abA | 61.4 ± 8.8 bA |
| MNPK | 41.6 ± 11.9 aB | 108.6 ± 11.5 aB | 27.1 ± 5.1 aA | 282.2 ± 34.7 aB | 8.6 ± 2.1 aB | 68.4 ± 5.1 aB | 78.1 ± 9.9 aB | 15.0 ± 5.1 aB | 45.7 ± 7.7 aB | 118.6 ± 9.9 aA | 269.6 ± 32.1 aA | 19.9 ± 3.6 faA | 466.0 ± 34.8 aA | 31.4 ± 7.6 aA | 148.2 ± 9.9 aA | 139.7 ± 10.7 aA | 28.3 ± 6.4 aA | 101.4 ± 8.9 aA |

| Treatment | ΔResin-P (%) | ΔNaHCO$_3$-P$_i$ (%) | ΔNaHCO$_3$-P$_o$ (%) | ΔNaOH-P$_i$ (%) | ΔNaOH-P$_o$ (%) | ΔDHCl-P$_i$ (%) | ΔCHCl-P$_i$ (%) | ΔCHCl-P$_o$ (%) | ΔResidual-P (%) | ΔResin-P % (%) | ΔNaHCO$_3$-P$_i$ (%) | ΔNaHCO$_3$-P$_o$ (%) | ΔNaOH-P$_i$ (%) | ΔNaOH-P$_o$ (%) | ΔDHCl-P$_i$ (%) | ΔCHCl-P$_i$ (%) | ΔCHCl-P$_o$ (%) | ΔResidual-P (%) |
|---|---|---|---|---|---|---|---|---|---|---|---|---|---|---|---|---|---|---|
| NPK | 0.2 ± 0.04 bB | 7.3 ± 2.0 aB | 0.2 ± 0.02 cB | 6.9 ± 1.1 cA | −1.9 ± 0.4 aA | 0.1 ± 0.02 cB | −5.6 ± 0.7 aA | −0.1 ± 0.02 aA | −7.2 ± 1.1 adA | 8.9 ± 1.1 aA | 12.1 ± 2.1 aA | 0.4 ± 0.1 aA | 3.6 ± 0.7 cB | −2.6 ± 0.8 aA | 3.6 ± 0.3 bA | −9.9 ± 1.0 aB | −0.23 ± 0.1 bB | −15.8 ± 4.7 aB |
| NPKS | 2.2 ± 0.08 aB | 4.8 ± 0.8 bB | 0.9 ± 0.2 bA | 7.6 ± 1.2 cA | −2.6 ± 0.5 bA | 0.5 ± 0.03 bB | −5.6 ± 0.9 aA | −0.6 ± 0.04 bB | −7.2 ± 2.0 aA | 9.4 ± 2.5 aA | 10.0 ± 1.8 aA | −0.02 ± 0.01 cB | 6.9 ± 0.8 bA | −4.6 ± 0.9 bB | 3.2 ± 0.4 bA | −9.0 ± 1.3 aB | 0.02 ± 0.01 aA | −16.0 ± 1.9 aB |
| M | 2.7 ± 0.9 aB | 7.0 ± 1.1 aB | 1.5 ± 0.1 aA | 9.8 ± 0.7 bB | −3.3 ± 1.0 bB | 4.0 ± 0.7 aA | −11.1 ± 1.0 bA | −1.4 ± 0.3 cB | −9.2 ± 2.4 abA | 6.1 ± 1.6 aA | 12.7 ± 1.7 aA | 0.01 ± 0.02 bB | 13.6 ± 1.2 aA | −3.2 ± 0.6 abA | 4.4 ± 1.3 abA | −16.1 ± 1.7 bB | −0.8 ± 0.1 cC | −16.8 ± 2.2 aB |
| MNPK | 2.9 ± 0.8 aB | 8.0 ± 0.9 aB | 1.4 ± 0.2 aA | 13.8 ± 0.9 aA | −3.6 ± 1.1 bA | 4.7 ± 1.0 aA | −14.5 ± 2.6 bA | −1.6 ± 0.3 cB | −11.1 ± 1.7 bA | 7.0 ± 0.9 aA | 14.1 ± 2.4 aA | 0.01 ± 0.01 bB | 13.9 ± 2.0 aA | −3.3 ± 1.1 abA | 5.8 ± 1.1 aA | −18.5 ± 2.8 bB | −1.0 ± 0.2 cC | −17.0 ± 2.8 aB |

[1] ΔResin-P, etc: Resin-P in 1995 minus that in 2000. Similarly, it is as same for other P fractions for the Hedley Method in the year 2013. [2] NPK, nitrogen, phosphorus, and potassium; NPKS, nitrogen, phosphorus, potassium, and straw; M, manure; MNPK, manure, nitrogen, phosphorus, and potassium. Factor levels marked with the same letter do not differ at the $p < 0.05$ level of significance. Lowercase letters (a,b,c) indicate significant difference of P fractions in treatments, and the uppercase letters (A,B,C) indicate significant difference of P fractions in years.

**Table 4.** Increase amount and percentage of P fractions for the JiangGu Method.

| Treatment | 2000 (year) | | | | | | 2013 (year) | | | | | |
|---|---|---|---|---|---|---|---|---|---|---|---|---|
| | $\Delta Ca_2$-P [1] (mg kg$^{-1}$) | $\Delta Ca_8$-P (mg kg$^{-1}$) | $\Delta Al$-P (mg kg$^{-1}$) | $\Delta Fe$-P (mg kg$^{-1}$) | $\Delta O$-P (mg kg$^{-1}$) | $\Delta Ca_{10}$-P (mg kg$^{-1}$) | $\Delta Ca_2$-P (mg kg$^{-1}$) | $\Delta Ca_8$-P (mg kg$^{-1}$) | $\Delta Al$-P (mg kg$^{-1}$) | $\Delta Fe$-P (mg kg$^{-1}$) | $\Delta O$-P (mg kg$^{-1}$) | $\Delta Ca_{10}$-P (mg kg$^{-1}$) |
| NPK [2] | 2.3 ± 1.0 bB | 6.1 ± 4.2 cB | 47.5 ± 10.7 cB | 344.4 ± 94.1 bA | 8.4 ± 2.9 aA | 6.6 ± 3.6 aB | 88.7 ± 26.3 cA | 73.9 ± 11.1 cA | 96.0 ± 13.2 aA | 90.5 ± 33.6 bB | 65.1 ± 13.6 bA | 31.9 ± 5.6 bA |
| NPKS | 3.3 ± 0.4 bB | 14.3 ± 1.1 bB | 64.2 ± 6.0 bB | 348.4 ± 70.4 bA | −5.3 ± 0.8 cB | −4.5 ± 0.9 cB | 98.4 ± 23.3 cA | 84.5 ± 20.0 cA | 102.4 ± 20.8 aA | 100.7 ± 8.5 bB | 67.5 ± 10.1 bA | 30.7 ± 8.3 bA |
| M | 58.8 ± 21.0 aB | 63.6 ± 27.8 aB | 114.6 ± 36.1 aA | 552.0 ± 119.5 aA | 0.2 ± 0.5 bB | 8.5 ± 2.4 bB | 145.1 ± 21.2 bA | 130.6 ± 23.5 bA | 127.4 ± 34.4 aA | 154.7 ± 24.8 aB | 102.8 ± 6.9 aA | 69.9 ± 5.8 aA |
| MNPK | 69.5 ± 20.1 aB | 68.3 ± 21.7 aB | 151.5 ± 45.2 aA | 707.8 ± 161.2 aA | −5.9 ± 0.8 cB | 16.9 ± 4.3 aB | 205.6 ± 16.1 aA | 250.9 ± 33.3 aA | 136.1 ± 23.0 aA | 160.9 ± 24.2 aB | 104.4 ± 11.9 aA | 76.1 ± 10.7 dA |
| Treatment | $\Delta Ca_2$-P (%) | $\Delta Ca_8$-P (%) | $\Delta Al$-P (%) | $\Delta Fe$-P (%) | $\Delta O$-P (%) | $\Delta Ca_{10}$-P (%) | $\Delta Ca_2$-P (%) | $\Delta Ca_8$-P (%) | $\Delta Al$-P (%) | $\Delta Fe$-P (%) | $\Delta O$-P (%) | $\Delta Ca_{10}$-P (%) |
| NPK | −2.0 ± 1.5 bB | −0.3 ± 0.4 bB | 0.9 ± 0.3 bA | 32.3 ± 12.3 aA | −27.8 ± 9.5 aB | −3.1 ± 1.1 aB | 18.0 ± 8.1 aA | 16.6 ± 6.0 aA | 13.0 ± 33 aA | 6.2 ± 2.2 aB | −46.2 ± 9.3 aA | −3.2 ± 0.4 abA |
| NPKS | −2.3 ± 0.9 bB | −0.5 ± 0.7 bB | 1.0 ± 0.3 bA | 32.2 ± 7.7 aA | −26.3 ± 4.8 aB | −4.1 ± 0.9 aB | 18.4 ± 7.2 aA | 14.3 ± 4.4 aA | 15.7 ± 9 aA | 5.1 ± 1.1 aB | −45.9 ± 11.8 aA | −4.1 ± 1.0 bA |
| M | 1.8 ± 0.5 aA | 4.8 ± 2.0 aA | 4.8 ± 1.2 aA | 23.9 ± 7.7 aA | −31.0 ± 10.7 aB | −4.2 ± 2.1 aB | 16.5 ± 4.9 aB | 15.1 ± 7.4 aB | 11.3 ± 5.6 aB | 5.5 ± 0.9 aB | −38.2 ± 5.7 aA | −8.2 ± 0.9 cA |
| MNPK | 1.8 ± 0.5 aA | 3.6 ± 1.2 aA | 5.4 ± 2.2 aA | 26.4 ± 10.8 aA | −33.0 ± 8.8 aB | −4.1 ± 2.8 aB | 18.2 ± 4.3 aB | 24.3 ± 8.8 aB | 7.4 ± 2.2 aB | 1.8 ± 0.6 bB | −49.7 ± 9.0 aA | −2.0 ± 0.5 aA |

[1] $\Delta Ca_2$-P, etc: $Ca_2$-P in 1995 minus that in 2000. Similarly, it is as same for other phosphorus fractions for the Jiang Gu method in the year 2013. [2] NPK, nitrogen, phosphorus, and potassium; NPKS, nitrogen, phosphorus, potassium, and straw; M, manure; MNPK, manure, nitrogen, phosphorus, and potassium. Factor levels marked with the same letter do not differ at the $p < 0.05$ level of significance. Lowercase letters (a,b,c) indicate significant difference of P fractions in treatments, and the uppercase letters (A,B,C) indicate significant difference of P fractions in years.

Compared to single application of chemical fertilizers, the combination of manure and straw significantly increased the P pool ($p < 0.05$). After 10 years of fertilization, the labile P (Resin-P + NaHCO$_3$-P) increased by 76.9 mg kg$^{-1}$ for NPKS treatment, and the MNPK treatment increased it by 177.3 mg kg$^{-1}$, which was 2.3 times higher than the NPKS treatment. NPKS and MNPK treatment increased the labile P ratio by 7.9% and 12.3%. Similarly, the increase in medium labile P (NaOH-P + DHCl-P$_i$) with MNPK treatment was 4.1 times higher than NPKS treatment. After 23 years of fertilization, the labile P increased by 260.4 mg kg$^{-1}$ for NPKS treatment, and 408.1 mg kg$^{-1}$ for MNPK treatment, which was 1.6 times higher than the NPKS treatment. In summary, the application of P fertilizer increased the labile phosphorus and medium labile P, and decreased the proportion of stable P. The increase in M and MNPK treatments was higher than that in NPK and NPKS treatments.

### 3.5. Influence Factors of ΔOlsen P and ΔPAC

Different fertilization methods affected Olsen P, PAC, P fractions, as well as soil properties, and there was correlation between them. In order to clarify the correlation between ΔOlsen P, ΔPAC and P fractions, as well as soil factors, we chose RDA analysis to analyze its relevance (Figure 4). The interpretation rate of soil P fractions increment for Olsen P and PAC was 95.1% in Figure 4a. The first redundancy factor (RDA1) explained 67.7% of the variation, and the second redundancy factor (RDA2) explained 27.4% of the variation. According to the interactive forward selection method, Al-P, Ca$_2$-P, Resin-P, and NaOH-P$_i$ were the main factors affecting soil P adsorption–desorption, and the interpretation rate was 48.1% ($p < 0.05$). With ΔOlsen P, ΔPAC, and soil properties in Figure 4b, the interpretation rate of soil properties for Olsen P and PAC was 89%, the first redundancy factor RDA1 explained 81.5% of the variation, and the second redundancy factor RDA2 explained 7.5% of the variation. According to the interactive forward selection method, C/N, SOC, and pH were the main factors affecting soil Olsen P and PAC, and the interpretation rate was 45.8% ($p < 0.05$). In conclusion, Al-P, Ca$_2$-P, Resin-P, and NaOH-P$_i$ in P fractions and C/N, SOC, and pH in soil properties had a great influence on ΔOlsen P and ΔPAC. Fertilization affected the chemical properties of red soil and also affected the transformation of the P pool. In the case of P surplus, the interaction of C/N, SOC, and pH, in addition to Olsen P and PAC was greater.

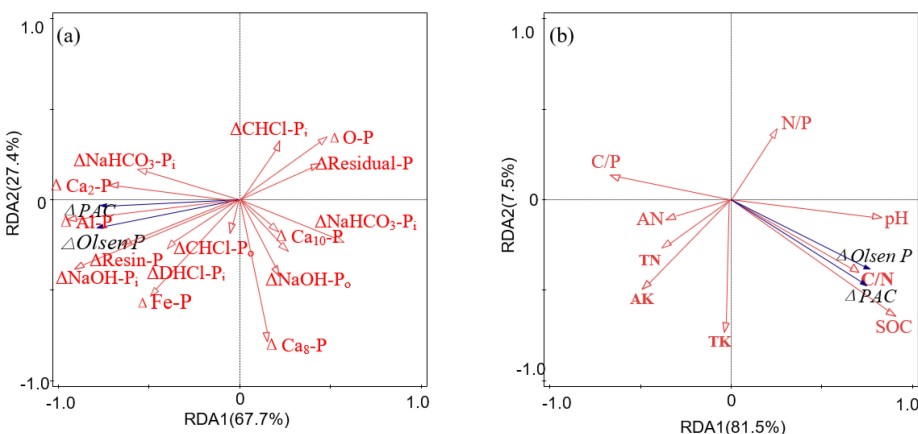

**Figure 4.** Influence factors of Olsen P and PAC increment (RDA). (**a**) RDA analysis of ΔOlsen P, ΔPAC, and change amount of P fractions. (**b**) RDA analysis of ΔOlsen P, ΔPAC, and chemical properties of soil.

## 4. Discussion
### 4.1. ΔOlsen P, ΔPAC, and P surplus

Different fertilizer inputs lead to different trends in the increase of soil P accumulation. This study showed that compared to straw application, manure application significantly

increased the accumulation of soil P. At the same time as the increase in the accumulation of P, Olsen P and PAC showed an increasing trend. NPK and NPKS treatments were similar and increased linearly, while M and MNPK treatments were similar and increased in an S-curve, and there was a period of rapid growth and equilibrium. It showed that manure significantly affected the increase of Olsen P and PAC in red soil.

The different increasing trends of Olsen P and PAC in red soil may be due to the following two reasons. Firstly, the amount of TP in fertilizer is different, and the amount of P in manure is higher than that in chemical fertilizer. The increase in input caused the increase in accumulated P, and the increasing trends of Olsen P and PAC were different under the conditions of high accumulation and low accumulation. Withers' research on farmland in the UK showed that the application of fertilizers and manure lead to the accumulation of TP and easily exchangeable P. The cumulative P surplus was influenced by the amount of P input [28]. Previous research showed that the increase in the range of Olsen P and PAC was large for a high input amount of P, and the input amount of P determines its increasing trend [29,30]. Guo's research showed that both manure and straw returning to the field could increase soil Olsen P, microbial biomass P, and acid phosphatase activity during a 110 day cultivation period [31]. Secondly, Olsen P and PAC treated by M and MNPK did not increase continuously but reached equilibrium in the later stage. The possible reason is that the high accumulation of soil P leads to its leaching and infiltration. Hua's research showed that the application of pig manure increased the leaching risk of dissolved unreactive phosphorus (DUP), while the application of straw can reduce the loss of DUP. Compared with NPK, the annual loss of DUP significantly increased by 11.1% and 8.6% in the treatment of single application of manure and manure combined with NPK, respectively, while it decreased by 6.2% in the treatment of NPK combined with straw [32]. Many studies believed that continuous and excessive P input would cause P infiltration and loss, especially in areas with heavy rainfall [33–35]. Table S2 of the supplementary provides a table of changes in the Olsen P profile; high P input increases the infiltration of Olsen P. Therefore, P loss should be paid attention to in long-term manure application.

### 4.2. Change of P Fractions

Long-term different carbon source inputs had a significant impact on the P fractions. The soil has a strong adsorption and fixation effect on P, especially in the red soil area where the Fe-Al oxide levels are high, which is also one of the important reasons for the low utilization rate of P fertilizer [33,36,37]. This study indicated that there were differences in the effects of manure and straw on P fractions, which may be mainly due to the following two factors: 1. Different types and inputs of P fertilizers result in different increases in P fractions. Lan and Li 's study showed that the increase in soil P application could significantly increase P pool and P fractions. The application of straw combined with chemical fertilizer P increased the accumulation of inorganic and organic P that could be extracted from $NaHCO_3$ and NaOH, and straw could partially replace the input of fertilizer P, reducing soil P accumulation and environmental risks [34,35]. Soma's study showed that the P input of manure was higher; manure significantly increased organic P and Resin-P by 35% and 64% after 10 and 32 years of sorghum cultivation, respectively, and significantly increased HCl-P after 32 years of cultivation. After 32 years of long-term manure and straw returning to the field, manure significantly increased $NaHCO_3$-$P_i$ and NaOH-$P_i$ by 63% and 26%, respectively. Manure had the best effect on increasing sorghum yield, which was closely related to soil pH, carbon, and nitrogen [38]. 2. The interaction between soil properties and P fertilizers. Both Bera and Arias showed that soils with higher iron and aluminum oxide had a higher P adsorption capacity, and the P adsorption data conformed to the Langmuir equation [39,40]. The Langmuir equation was similar to the increasing curve of Olsen P and PAC treated by M and MNPK treatments in this paper, but the increasing trend was different from that treated by NPK and NPKS treatments. Our research showed that the increase of Olsen P will be fast at first and then slow only after the accumulated amount of P in the soil reaches a certain level.

*4.3. Influencing Factors of ΔOlsen P and ΔPAC*

Fertilization affected soil P transformation and soil chemical properties. RDA analysis of Olsen P, PAC, and soil chemical properties showed that the contribution rate of soil C/N, SOC, and pH to Olsen P and PAC in accumulated P was the highest, which was similar to many previous research results. The main reasons may be due to the following two aspects: 1. There was an interaction between soil carbon, nitrogen, P, and pH. Miller's study on long-term soil fertilization for 9 years showed that both straw and manure increased the accumulation of soil P, and the main reason for the difference in their increase was the carbon nitrogen ratio of the soil [41]. Especially, the different carbon sources have a significant impact on the activity of P. Zhan's research on black soil also showed that soil organic matter and pH had an important impact on the change of soil Olsen P, and under the balance of 100 kg ha$^{-1}$ P, Olsen P and PAC were significantly correlated with soil organic matter content [42]. Yang's research showed that when the organic matter was 75.3 g kg$^{-1}$, the P efficiency was the highest, and that the addition of manure improves the availability of P by reducing the adsorption strength and maximum P release capacity of P [25].2. The imbalanced input of carbon and nitrogen affects the absorption and conversion of P. Wang's research showed that straw retention in farmland reduces nitrogen leaching loss due to its high carbon to nitrogen ratio, which enhances microbial nitrogen fixation [43]. In addition, Gérard showed that clay minerals are important binding sites of P, and their P control amount may exceed that of Fe-Al oxides [44]. Lü's research has shown that soil pH, organic matter, and C/N were related to changes in P fractions [45]. Cao's research showed that compared with NPK, NPKS treatment increased the proportion of NaHCO$_3$-P$_o$ and NaOH-P$_i$, and decreased the proportion of DHCl-P. Furthermore, Residual-P, soil SOC, and C/P had a significant impact on it [22]. Fang reported that particle morphology plays an important role in the adsorption of the solid–water interface, and the distribution of surface charges and reaction sites will affect the adsorption capacity of P [46]. These results are similar to this study; however, this paper cannot explain how SOC and pH affect Olsen P and PAC. The complex surface morphology may be important in particle adsorption. Therefore, whether SOC and pH affect Olsen P by affecting the distribution of soil particles and charges needs further study.

## 5. Conclusions

After 28 years of different fertilization methods, P application affected the increasing trend of Olsen P and PAC in red soil P accumulation. Olsen P and PAC of M and MNPK treatments increased in stages; Olsen P balance values were 167 and 164 mg kg$^{-1}$ and PAC equilibrium values were 10.4 and 10.2 mg kg$^{-1}$, respectively. Therefore, the results indicated that manure contributed more to the increase of P than straw returning in red soil. Therefore, attention should be paid to the risk of P loss in the long-term application of manure. Long-term application of P had different effects on Al-P and Fe-P. Al-P continued to increase, while Fe-P increased rapidly in the early stage. Labile P and medium labile P in P fractions, C/N, SOC, and pH in soil properties had the highest interpretation rates for Olsen P and PAC, with 48.1% and 45.8%, respectively. Therefore, there were differences in the effects of manure and straw on P fractions and soil properties, leading to different trends in Olsen P and PAC. Our research results filled the knowledge gaps of the increasing trend of Olsen P and PAC in accumulated P under manure and straw application, and clarified the changing trend of P fractions and soil properties, which was very important for formulating fertilization strategies for red soil. In the future, we would like to know more about the impact of the types and inputs of manure and straw on the increase of Olsen P in red soil, and more about the impact of manure and straw components on P fixation and loss, so as to provide better data support for P fertilizer application and policy formulation.

**Supplementary Materials:** The following supporting information can be downloaded at: https://www.mdpi.com/article/10.3390/agronomy13061647/s1, Table S1: Nutrient Content of Manure and Straw; Table S2: Soil Olsen P Profile in 2001 (mg kg$^{-1}$), Table S3. crop yield (kg ha$^{-1}$), Table S4. physical properties and chemical properties (2018) in red soil.

**Author Contributions:** Conceptualization: F.S. and M.X.; Methodology: M.X. and N.S.; Validation: F.S. and M.X.; Formal analysis: F.S.; Investigation: F.S. and M.X.; Resources: M.X., B.W., M.X. and Z.C.; Data curation: M.X.; Writing original draft: F.S.; Writing-review and editing: M.X. and N.S.; Supervision: M.X.; Project administration: N.S.; Funding acquisition: N.S. All authors have read and agreed to the published version of the manuscript.

**Funding:** This study was funded by the National Key Research and Development Program of China (2021YFD1901205) and the National Natural Science Foundation of China (42177341). We are very grateful to all colleagues for their efforts working on Qiyang long-term experiment.

**Data Availability Statement:** Not applicable.

**Conflicts of Interest:** The authors declare no conflict of interest.

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
