# Peer review of "Significant Effects of Long-Term Application of Straw and Manure Combined with NPK Fertilizers on Olsen P and PAC in Red Soil"

_agronomy, doi:10.3390/agronomy13061647_

Round 1
Reviewer 1 Report
The manuscript needs minor revision for improvement.
use P in the text not phosphorus every where
Check English
see the attached file for more comments
most of the references in the list are old, add new references (2020, 2021, 2022, 2023)

Minor English Check
Author Response
Dear Editor and Reviewer:
Thank you for your letter and for the Reviewer’s comments concerning our manuscript entitled “Long term application of straw and manure significantly changed the Olsen P and phosphorus activation coefficient in red soil of Southern China (2436632)”. Thank you very much for your careful reading of the whole article and your valuable suggestions for revision. Those comments are all valuable and helpful for improving our paper, as well as the important guiding significance to our researchers. I tried my best to revise and improve this article carefully which we hope meet with approval. Revised portion are marked in red in the paper. All revised lines and page numbers are corresponding manuscripts: "Revised manuscript-2436632".
The main corrections in the paper and the responds to the reviewer’s comments are as follows:
Reply to the Review Report 1
Reviewer’s comments:
(1)Use P in the text not phosphorus every where.
(2)Check English.
(3)Most of the references in the list are old, add new references (2020, 2021, 2022, 2023).
(4)The attached file comments.
Author’s response: Thank you for your suggestion. We checked the entire article and replaced phosphorus with P. “hm-2” is revised to “ha-1” throughout the entire article.
Title: Based on the suggestion of another reviewer, the title is revised to: Significant Effects of Long-Term Application of Straw and Manure Combined with NPK Fertilizers on Olsen P and PAC in Red
Abstract:
The abstract has been rewritten, with modifications including:
(1) The already explained words (N, P, K, M, S) have been abbreviated for the second time.
(2) Refine the language, with a total word count of no more than 200 words in the abstract.
(3) Uncertain descriptions have been removed.
Introduction:
(1) Replaced phosphorus with P.
(2) Reference description and citation errors, according to the request of another reviewer, the introduction has been revised and all similar issues have been corrected.
(3) “First, Third” is revised to “Firstly, Thirdly”. (Line 202, 206, page5)
Materials and methods:
There was a description error. “a total of 52.4 kg hm−2 yr−1 of fertilizer phosphorus was applied to wheat and maize every year” is revised to “a total of 52.4 kg ha−1 of fertilizer P was applied to wheat and maize every year” (Line 147, page 6); “45 kg hm−2 of carbofuran was sprayed” is revised to “4.5 kg hm−2 of carbofuran was sprayed” . (Line 261, page 6)
Results:
(1) There was a Punctuation error. “. (Figure 2b).” is revised to “(Figure 2b).” (Line 332, page 8); “(2000 and 2013) respectively” is revised to“(2000 and 2013), respectively” . (Line 147, page 6)
(2) An error occurred in the description. “After 10 years of fertilization (2000 year)” is revised to “After 10 years of fertilization (2000)” . (Line 362, page 9)
Discussion:
“There are two possible reasons” is revised to “The different increasing trends of Olsen P and PAC in red soil may be due to the following two reasons.” (Line 362, page 9)
References:
Add 7 new references (2020, 2021, 2022, 2023).
- Zhang, W.W.; Wang, Q.; Wu, Q.H.; Zhang, S.X.; Zhu, P.; Peng, C.; Huang, S.M.; Wang, B.R.; Zhang, H.M. The response of soil Olsen-P to the P budgets of three typical cropland soil types under long-term fertilization. Plos one, 2020, 15 (3): e0230178. (Line 636-637)
- Cao, N., Zhi, M.L., Zhao, W.P., Pang, J.Y., Hu, W., Zhou, Z.G., Meng, Y.L. Straw retention combined with phosphorus fertilizer promotes soil phosphorus availability by enhancing soil P-related enzymes and the abundance of phoC and phoD genes. Soil and Tillage Research, 2022, 220, 105390. (Line 620-622)
- Chen, X.H., Yan, X.J., Wang, M.K., Cai, Y.Y., Weng, X.F., Su, D., Guo, G.X., Wang, W.Q., Hou, Y., Ye, D.L., Zhang, S.W., Liu, D.H., Tong, L., Xu, X.Z., Zhou, S.G., Wu, L.Q., Zhang, F.S. Long-term excessive phosphorus fertilization alters soil phosphorus fractions in the acidic soil of pomelo orchards. Soil and Tillage Research, 2022, 215, 105214. (Line 583-585)
- Fan, L., Zhao, T., Tarin, M. W. K., Han, Y.Z., Hu, W.F., Rong, J.D., He, T.Y., Zheng, Y.S. Effect of various mulch materials on chemical properties of soil, leaves and shoot characteristics in Dendrocalamus Latiflorus Munro forests. Plants, 2021, 10(11), 2302. (Line 576-578)
- Battisti, M., Moretti, B., Sacco, D., Grignani, C., Zavattaro, L. Soil Olsen P response to different phosphorus fertilization strategies in long-term experiments in NW Italy. Soil use and management, 2022, 38:549-563. (Line 623-624)
- Tandy, S., Hawkins, J.M.B., Dunham, S.J., Hernandez‐Allica, J., Granger, S. J., Yuan, H.M., McGrath, S.P., Blackwell, M. S. Investigation of the soil properties that affect Olsen P critical values in different soil types and impact on P fertiliser recommendations. European Journal of Soil Science, 2021, 72(4), 1802-1816. (Line 617-619)
- Demay, J., Ringeval, B., Pellerin, S., Nesme, T. Half of global agricultural soil phosphorus fertility derived from anthropogenic sources. Nature Geoscience, 2023, 16, 69-74. (Line 569-570)
Reviewer 2 Report
I have finished my review on the Manuscript Number: agronomy-2436632 Title: Long term application of straw and manure significantly changed the Olsen P and phosphorus activation coefficient in red soil of Southern China.
1. Generally, the manuscript is well written and presents an interesting topic for the journal of Agronomy.
2. Please write accurate keywords. Like the red soil and straw, these are not appropriate for this paper?
3. The Abstract needs to be rewritten, especially in the methods and objectives sections.
4. Line 43-45, You need to have logical reasons for increasing yield. This can be due to the effects of mulch on many physical and chemical properties of the soil. Please read and add references as follows:
Fan, L.; Zhao, T.; Tarin, M.W.K.; Han, Y.; Hu, W.; Rong, J.; He, T.; Zheng, Y. Effect of Various Mulch Materials on Chemical Properties of Soil, Leaves and Shoot Characteristics in Dendrocalamus Latiflorus Munro Forests. Plants 2021, 10, 2302.
Hongyan Cheng, Xiaozhen Zhu….. Effects of different mulching and fertilization on phosphorus transformation in upland farmland. Journal of Environmental Management.2020.
Misagh Parhizkar, Mahmood Shabanpour ……Effects of length and application rate of rice straw mulch on surface runoff and soil loss under laboratory simulated rainfall. International Journal of Sediment Research 36 (2021) 468e478.
M. G. Mostofa Amin, Ahmed Al Minhaj….. Mulch and no-till impacts on nitrogen and phosphorus leaching in a maize field under sub-tropic monsoon climate. Environmental Challenges. Volume 5, December 2021, 100346
Ni X, Song W, Zhang H, Yang X, Wang L. Effects of Mulching on Soil Properties and Growth of Tea Olive (Osmanthus fragrans). PLoS One. 2016 Aug 10; 11(8):e0158228.
5. Line 136, please add figure showing the geographical location and aerial map for your study site.
6. why did you choose this study area? According to what? Investigation or other researches?
7. The results and Discussion sections are OK.
Minor editing of English language required
Author Response
Dear Editor and Reviewer:
Thank you for your letter and for the Reviewer’s comments concerning our manuscript entitled “Long term application of straw and manure significantly changed the Olsen P and phosphorus activation coefficient in red soil of Southern China (2436632)”. Thank you very much for your careful reading of the whole article and your valuable suggestions for revision. Those comments are all valuable and helpful for improving our paper, as well as the important guiding significance to our researchers. I tried my best to revise and improve this article carefully which we hope meet with approval. Revised portion are marked in red in the paper. All revised lines and page numbers are corresponding manuscripts: "Revised manuscript-2436632".
The main corrections in the paper and the responds to the reviewer’s comments are as follows:
Reply to the Review Report 2
Reviewer’s comments: Please write accurate keywords. Like the red soil and straw, these are not appropriate for this paper?
Author’s response: The keywords have been modified.
Keywords: red soil; manure; straw; Olsen P; phosphorus activation coefficient
Reviewer’s comments: The Abstract needs to be rewritten, especially in the methods and objectives sections.
Author’s response: The abstract has been rewritten, and another reviewer suggested that the abstract should be within 200 words, so the word count is limited.
Abstract: Application of manure (M) and straw (S) will increase Olsen P and phosphorus activation coeffi-cient (PAC) in soil, clarifying the increasing trend of Olsen P and PAC is crucial for rational ferti-lization. This study fitted the equation between the accumulated P surplus, Olsen P and PAC in four treatments for 28 years, analyzed the changes and rates of P fractions. The results showed Olsen P and PAC increase linearly with NPK and NPKS treatments, for every 100 kg ha−1 of P surplus, Olsen P increased by 5.9 and 6.7 mg kg−1, PAC increased by 0.52% and 0.50%. With M and MNPK treatments, the sigmoid curve equation was the best fitting method. The equi-librium values were 167 and 164 mg kg−1 for Olsen P, and 10.4 and 10.2 mg kg−1 for PAC. There was a correlation be-tween Al-P, Ca2-P, Resin-P, NaOH-Pi, C/N, SOC and pH had the highest interpretation rates for Olsen P and PAC. Manure is significantly higher than straw to improve Olsen P in red soil , it is recommended to reduce the amount of manure applied for a long time to avoid a zero increase in Olsen P.
Reviewer’s comments: Line 43-45, You need to have logical reasons for increasing yield. This can be due to the effects of mulch on many physical and chemical properties of the soil.
Author’s response: Add references as follows:
Fan, L., Zhao, T., Tarin, M. W. K., Han, Y.Z., Hu, W.F., Rong, J.D., He, T.Y., Zheng, Y.S. Effect of various mulch materials on chemical properties of soil, leaves and shoot characteristics in Dendrocalamus Latiflorus Munro forests. Plants, 2021, 10 (11), 2302. (Line 576-578)
Reviewer 3 Report
Overall, this MS need to be improved. The Abstract need to summarize to 200 words, Introduction need to refine and summarize, Material and method need to re-write and re-arrange systematically. Additionally , the quality of graph and table need to be improved. The detail suggestion are listed in attached file

This MS need to be improved by proofers
Author Response
Dear Editor and Reviewer:
Thank you for your letter and for the Reviewer’s comments concerning our manuscript entitled “Long term application of straw and manure significantly changed the Olsen P and phosphorus activation coefficient in red soil of Southern China (2436632)”. Thank you very much for your careful reading of the whole article and your valuable suggestions for revision. Those comments are all valuable and helpful for improving our paper, as well as the important guiding significance to our researchers. I tried my best to revise and improve this article carefully which we hope meet with approval. Revised portion are marked in red in the paper. All revised lines and page numbers are corresponding manuscripts: "Revised manuscript-2436632".
The main corrections in the paper and the responds to the reviewer’s comments are as follows:
Reply to the Review Report 3
Reviewer’s comments: The title is too long, need to refine and reflect the whole contents of MS.
Author’s response: Thank you for your suggestion. Another reviewer suggested using abbreviations. So, based on the suggestion of reviewers, the title is revised to: Significant Effects of Long-Term Application of Straw and Manure Combined with NPK Fertilizers on Olsen P and PAC in Red Soil.
Reviewer’s comments: Abstract is too long. Rewrite/Rearrange and summarize the abstract (follow the guidance). Abstract maximum 200 words. Please follow the author guidance and write concisely: • Short Background (Main problem or the importance of this work). • Objective: The study was done to investigated. • Methods. and data collected • Results and main conclusions.
Author’s response: Thank you again for your detailed guidance. Based on the four aspects that explained, we has revised the abstract to 200 words.
Abstract: Application of manure (M) and straw (S) will increase Olsen P and phosphorus activation coeffi-cient (PAC) in soil, clarifying the increasing trend of Olsen P and PAC is crucial for rational ferti-lization. This study fitted the equation between the accumulated P surplus, Olsen P and PAC in four treatments for 28 years, analyzed the changes and rates of P fractions. The results showed Olsen P and PAC increase linearly with NPK and NPKS treatments, for every 100 kg ha−1 of P surplus, Olsen P increased by 5.9 and 6.7 mg kg−1, PAC increased by 0.52% and 0.50%. With M and MNPK treatments, the sigmoid curve equation was the best fitting method. The equi-librium values were 167 and 164 mg kg−1 for Olsen P, and 10.4 and 10.2 mg kg−1 for PAC. There was a correlation be-tween Al-P, Ca2-P, Resin-P, NaOH-Pi, C/N, SOC and pH had the highest interpretation rates for Olsen P and PAC. Manure is significantly higher than straw to improve Olsen P in red soil , it is recommended to reduce the amount of manure applied for a long time to avoid a zero increase in Olsen P.
Reviewer’s comments: Introduction is too long; Use international units.
Author’s response:
- Rewrite the introduction, refine the sentences, and modify the sentences that repeat descriptions.
- Changed hm-2 to ha-1 throughout the article, including figures and tables.
Reviewer’s comments: Need to be re-written and rearranged Material & Methods systematically and concisely.
Re-write and Rearrange systematically
2.1. Site description and Used Material
- Explain briefly the wheat maize rotation system and its productivity
- Soil Properties (chemical and physical properties
- Pig manure and straw chemical composition. Explain in detail and the total amount applied yearly (use international unit ton ha-1 yr-1)
(2)2.2. Experimental Design and Field Management
Author’s response:
Thank you again for your suggestion. We have attached the additional tables for supplementary Material (supplementary table).
(1) Added crop yield table in supplementary Material (supplementary table 3) and added textual description.
The added description section is: “Adopting a double cropping rotation system of wheat and corn, the fertilizer application in the corn season accounts for 70% of the manure application, while wheat accounts for 30%. The fertilizer is applied as a base fertilizer before sowing wheat and corn. Except for NPKS treatment where half of the crop straw is returned to the field, all the aboveground parts of the other treated crops are taken away”. (Line 250-256)
The yield of wheat and corn from the initial period of 1990 to 2018 is shown in supplementary table 3.
(2) Added the chemical and physical properties of the soil in 2018. The table of soil physicochemical properties for the four treatments in 2018 is attached in supplementary table 4. The physical and chemical properties of the initial soil are shown in Table 1. (Line 223-230)
(3) The annual input of pig manure is the same in every year, and half of the straw is returned to the field. Added descriptions of this section.
The input amount and water content of pig manure and straw from 1991, 2004, 2008 and 2012 were organized, and the content of carbon, nitrogen, phosphorus, and potassium was provided. This part is shown in Table 1 of supplementary. (Line 218-222)